# Knowledge and Practices of Four Onchocerciasis-Endemic Communities in Cameroon

**DOI:** 10.3390/microorganisms13040736

**Published:** 2025-03-25

**Authors:** Kamtsap Pierre, Nguemaïm Ngoufo Flore, Paguem Archile, Renz Alfons

**Affiliations:** 1Institute for Evolution and Ecology, Department of Comparative Zoology, University of Tübingen, Auf der Morgenstelle 28, 72076 Tübingen, Germany; alfons.renz@uni-tuebingen.de; 2Programme Onchocercoses Field Station, University of Tübingen, Ngaoundéré Box 65, Cameroon; 3Faculty of Health Sciences, University of Bamenda, Bambili P.O. Box 69, Cameroon; ngflorema@yahoo.fr; 4Faculty of Agriculture and Veterinary Medicine, Department Veterinary Medicine, University of Buea, Buea P.O. Box 63, Cameroon

**Keywords:** onchocerciasis, black fly, knowledge, practice, Cameroon, Befang, control, Soramboum, Mawong, Galim

## Abstract

This study examines the knowledge and practices of four onchocerciasis-endemic communities in Cameroon regarding black flies and the disease. Onchocerciasis, or river blindness, is caused by *Onchocerca volvulus* and transmitted by *Simulium* black flies. Despite over 20 years of ivermectin distribution, no vector control has been implemented, leading to high black fly densities. A survey of 452 individuals from Mawong, Menchum, Soramboum, and Galim revealed significant knowledge gaps. While 90% recognized black flies as a nuisance, only 9.3% knew their bites could cause blindness, and many mistakenly associated them with malaria. About 34.1% correctly identified evening as the main biting period, but misconceptions about breeding sites and transmission were common. Most respondents were unaware of precautionary measures beyond wearing appropriate gear. Misconceptions differed by location, depending on the education level and occupation. The study emphasizes the need for better community education on vector ecology, disease transmission, and prevention. Integrating targeted health education with ivermectin delivery, as well as implementing vector control measures, could help to eliminate onchocerciasis. Future studies should broaden geographical scope and include qualitative methods to better understand community attitudes and improve intervention techniques.

## 1. Introduction

Onchocerciasis or ‘river blindness’ is a neglected tropical parasitic disease caused by *Onchocerca volvulus,* which is transmitted in Africa mainly by black fly members of the *Simulium damnosum* complex, which breed in rapid-flowing rivers and streams [1,2,3]. Adult female *O. volvulus* worms live in subcutaneous nodules for over a decade [1,4,5]. A female worm is capable of releasing 1300–1900 microfilariae per day during the intermittent periods of reproduction, and these microfilariae are mainly found in the skin and eyes [6,7], where they survive for months or even years, waiting for a female black fly to pick them up during its blood meal. The infection is responsible for severe itching and depigmentation of the skin, as it occurs in African rainforest areas, and profound impairment of visual acuity, often leading to irreversible blindness, which tends to predominate in African savannah [8,9,10].

Black flies of the *S. damnosum* complex are the main vectors of human onchocerciasis in Cameroon. At least four sub-species have been identified in the rainforest and savannah of Cameroon: *Simulium sirbanum,* a small pale species discovered only in the Sudan savannah; *S. damnosum sensu stricto*, savannah mainly, and now observed in 1986 in the forest near Kumba; *Simulium squamosum*, forest and Guinea savannah [11,12,13]. Several subspecies of *S. squamosum* have been identified, including dark species in forests and pale species in savannah. *Simulium mengense* occurs mostly in the forest; however, it has occasionally been observed in the savannah [11,14,15].

Onchocerciasis is caused by *Onchocerca volvulus*, a parasitic worm that lives for up to 14 years in the human body. It is spread through the bite of a blackfly of the *Simulium damnosum* species complex, which breeds in fast-flowing rivers and streams. When the fly bites, it deposits the larvae of the parasitic worm, which matures to adulthood and produces millions of tiny worms, called microfilaria. Adult’s flies emerge after 8–12 days and live for up to four weeks, during which they can cover hundreds of kilometers in flight. Each adult female worm, thin but more than 0.5 m in length, produces millions of microfilariae (microscopic larvae) that migrate throughout the body. After mating, the female blackfly seeks a blood meal and may ingest microfilariae if the meal is taken from a person infected with onchocerciasis. A few of these microfilariae may transform into infective larvae within the blackfly, which are then injected into the person from whom the next meal is taken and subsequently develop into adult parasites, thus completing the life cycle of the parasite [16,17].

One of the most effective public health campaigns for managing onchocerciasis is mass distribution (MDA) of ivermectin. By lowering the microfilarial load in the skin, ivermectin reduces both the degree of clinical symptoms—such as severe itching, skin depigmentation, and eye lesions and the possibility of transmission by black flies. Although MDA programs show remarkable effectiveness, community involvement and compliance are absolutely crucial for their success. Several studies have noted that even after years of mass ivermectin distribution, local populations may still have significant gaps in their understanding of the disease and its vectors [18,19]. For example, many communities in endemic areas continue to misattribute black fly bites to other conditions like malaria, or they may lack knowledge about the specific behavior and breeding sites of these vectors [20]. This disconnect between biomedical intervention and local perceptions can hinder the overall effectiveness of MDA efforts. An inadequate understanding of the vector’s role means that community members might not fully appreciate the importance of consistent drug uptake, nor might they adopt complementary practices that reduce exposure to black flies. Therefore, integrating targeted health education with MDA programs is crucial. Educational campaigns that explain the life cycle of *Onchocerca volvulus*, the role of black flies in its transmission, and the benefits of ivermectin can help bridge the gap between biomedical strategies and local knowledge. In turn, better-informed communities are more likely to participate fully in MDA and support additional vector control measures, ultimately contributing to a more sustainable reduction in onchocerciasis transmission.

Related filarial parasites may be present in cattle and other livestock in endemic areas in addition to humans. Cattle parasite *Onchocerca ochengi* is one well-researched example [21,22]. The same black fly vectors that transport *O. volvulus* also carry *O. ochengi*, despite the fact that the latter does not infect humans. Our knowledge of parasite development, vector competence, and host immune responses has improved as a result of comparative investigations between these parasites. Black flies’ eating habits can also be impacted by the presence of livestock in endemic areas. In certain situations, cattle may act as substitute hosts (a process called zooprophylaxis), which could deter flies from attacking people. This relationship is complicated, though, because in some ecological contexts, cattle may actually boost local black fly abundance by providing more blood supplies, which could raise the likelihood of transmission overall.

In 2016, we performed a study in the northwest region of Cameroon where 1491 *Simulium damnosum s.l.* flies were captured during 6 days of catching from two catching sites: one in Mawong village near river Mawong, a tributary of the Menchum River, and another at the village of Befang near the Menchum River at the sand-digging site close to the Menchum Falls. Of the 558 parous flies dissected, 9.50% were infected with *Onchocerca* larvae (Table 1).

In 2016, the annual biting rates at the savannah sites were approximately 16,000 flies/man/year at Soramboum and 13,500 flies/man/year at Galim [21]. In 21,897 flies dissected from these two catching sites, only a few infective stage larvae of *O. volvulus* were observed in Soramboum, whereas no cases were reported in Galim [21].

The surveillance of the infectivity status of *Simulium* vectors biting people along river systems is vital for monitoring the transmission of onchocerciasis [7,22]. Understanding community perceptions and concerns related to onchocerciasis vectors is essential for the global elimination of the disease [18,23]. Community involvement may be a key component of onchocerciasis control activities, especially in the pre-elimination stages of vector control. To achieve community participation and design socially and locally acceptable control strategies, health program planners and implementers need to be familiar with the knowledge of local people and their attitudes toward onchocerciasis [24].

Much attention has been paid to the mass administration of ivermectin; however, insufficient information is available concerning the knowledge and perception of local populations regarding onchocerciasis vectors in Cameroon. Therefore, we investigated the knowledge and practice of black fly vectors of onchocerciasis in people living along the Menchum Valley, Galim in the Guinea savannah, and Soramboum in the Sudan savannah in Cameroon after more than 20 years of mass drug distribution to optimize the control strategy.

## 2. Methods

### 2.1. Description of Study Areas and Populations

A community-based cross-sectional survey was conducted in four localities in Cameroon: Befang near the river Menchum (6°18′25′′ N, 10°01′01′′ E) and Mawong near its tributary Mawong (6°19′26′′ N, 10°00′13′′ E) in the Northwest Region, Soramboum near the river Vina du Nord (7°47′14′′ N, 15°0′22′′ E) in the North region, and Galim close to the Mayo Djouroum, a tributary of the river Vina du Sud (7°12′10″ N, 13°35′46″ E) in the Adamawa region (Figure 1). These localities are crossed by, or located near, fast-flowing rivers, which serve as breeding sites for black fly vectors of onchocerciasis. Humans mainly live in livestock farming, agriculture, and sand collection areas.

The Menchum Valley subdivision has a total area of 4469 square km, 48 inhabitants/km^2^ with a growth rate of 3% per year, and as of 2005, a total population of 161,998 [25]. The Menchum River drains this area west into Nigeria and joins the Benue River. Generally, tropical grassland vegetation is composed of spear grass with non-uniform soil. Numerous fast-running streams merge with the Menchum River, creating a dendritic network that forms the main tributary of Katsina-Ala in Nigeria. The climate is warm and moist, with heavy rains and high temperatures reaching 37 °C between March and April, as well as hot days and cold nights [25].

Soramboum, a former hyperendemic village for onchocerciasis, is situated close to the Vina du Nord River in the Sudan savannah and has been the site of ivermectin mass treatments conducted since 1987 [26]. Today, the village has approximately 1500 inhabitants (personal observation). The Vina du Nord River flows perennially, with a maximum water discharge of ca. 500 m^3^ per second during the height of the rainy season and 5 m^3^/s at the end of the dry season (at Touboro, ca. 50 km downstream of Soramboum) [27].

The village of Galim is located on the Adamawa Plateau in a Guinea savannah region 15 km southwest of Ngaoundéré, approximately 3 km from the large river Vina du Sud (mean annual discharge 37 m^3^/sec, 1050 m altitude) [27]. The Adamawa Plateau is necessary for livestock production in Cameroon, and agriculture is the main economic activity practiced by the inhabitants.

*Simulium damnosum sensu stricto* and *S. sirbanum* are the predominant vector species in the Vina du Nord River, whereas *S. squamosum* is the predominant vector species in the Vina du Sud River [28,29].

### 2.2. Sampling

#### 2.2.1. Calculation of Sample Size and Selection of Recruitment Sites

Insufficient information is available concerning the level of community knowledge and practices regarding onchocerciasis along the Menchum Valley and in the other localities visited. We hypothesized that at least 50% of the population in the target area would not have a good level of knowledge regarding the transmission of onchocerciasis. Accordingly, the sample size was considered to have a 95% confidence level, a 5% margin of error, and 50% accuracy. By assuming maximum variability (50%), we ensured that our sample size would be large enough to provide reliable and precise estimates of the population’s knowledge and practices regarding onchocerciasis. We then came out with *n* = 452.

These communities were selected because of the observed symptoms of onchocerciasis (Figure 2) and the geographical characteristics of these areas, which are savannah-type drained by rivers favorable for the development of black flies. The main activities of the residents are agriculture, sand digging, and cattle and sheep rearing, which are also favorable for black fly activities.

#### 2.2.2. Community Mobilization and Data Collection

Structured questionnaires (Appendix A) were prepared in English and French and thoroughly explained in the relevant local language. Participants in Menchum Valley were interviewed in ‘Pidjin’, an English dialect mainly used in the villages of Befang and Mawong. In Galim and Soramboum, Fulfulde was used in the northern part of Cameroon with the aid of local guides.

Information concerning sociodemographic characteristics (including age, sex, occupation, body shape, and educational level), risk factors for the transmission of onchocerciasis, and variables related to participants’ knowledge and practices were included in the questionnaires. Age was grouped into intervals of 10 years. The questions were closed, and the participants included in this study were born or had lived in the study areas for several years. Residents who refused to answer all the questions or did not understand the questions despite explanations were excluded, as were immigrants from other countries and non-residents within the household.

#### 2.2.3. Ethical Approval

Ethical clearance and approval were obtained from the Institutional Review Board of the University of Douala (CEIUD/371/01/2016/M). The administrative authorization was obtained from the public health authorities of Cameroon. Informed consent was obtained from all voluntarily involved individuals after they received detailed explanations of the study in their local language. Each participant agreed verbally, and participants aged < 18 years participated only when their parents agreed to participation.

#### 2.2.4. Statistical Analysis

Recorded data were transferred to SigmatPlot version 15.0 and analyzed according to the study objectives. Chi-square tests were used to compare categorical variables (sex, occupation, and knowledge of the different villages). Frequencies and percentages were used to summarize the data. Continuous variables (age) are described as the median and interquartile range (IQR).

## 3. Results

### 3.1. Sociodemographic Characteristics of Study Participants

A total of 452 individuals were included: 136 from Mawong Village near Mawong, 160 from Befang near Menchum, 88 from Soramboum near Vina du Nord, and 68 from Galim near Mayo Djouroum. More males (67.7%) (the majority coming from Befang) participated in the study than females (Table 2).

The age of the respondents ranged from 14 to 50 years. In Befang, 65.0% of the participants were under 20 years old, whereas 36.4% of the participants in Soramboum were between 40 and 50 years old, with the lowest quartile being 25 years old, the upper quartile being 45 years old, the IQR being 20 years old, and the majority falling within the age group under 20 years. In contrast, sand dredging was prominent in Befang (58.8%) but absent elsewhere, and Galim displayed greater occupational diversity, including motorcycle cab drivers, shepherds, and builders.

A chi-square test of independence was performed to examine the difference between the localities and the capability of respondents with knowledge of the vectors of onchocerciasis based on educational level. This educational levels varied markedly; Mawong and Befang had higher primary and secondary school attendance, whereas Galim and Soramboum had notable rates of illiteracy (55.9% and 31.8%, respectively). In Mawong, 16 individuals (11.8%) had attended university, while no university attendance was reported in the other sites. (Table 2). The difference between the variables was statistically significant (chi-square = 220.91, *p*-value < 0.001). A significant difference was observed among the age groups, in which participants less than 20 years old were from Menchum and were mostly sand diggers (chi-square = 467.06, *p*-value < 0.001; Table 2).

### 3.2. Knowledge Practice of Community Respondents Regarding Biting Activities of Black Flies

All respondents had some knowledge of the vector of onchocerciasis, with 34.1% assuming that it bites in the evening. However, 0.4% had yet to learn when it was usually a bite. Of the participants from Mawong, Befang, Galim, and Soramboum, 17.6%, 30.0%, 5.9%, and 11.4%, respectively, were confident that black flies would bite in the morning. Regarding the effect of bites on humans, 70.0% of Menchum and 64.7% of Mawong reported that black fly bites caused malaria, whereas 11.8% of Mawong reported that black fly bites caused blindness (Table 3).

More than 44% of the study participants thought that most black fly bites occurred near rivers, and 27.4% believed that farms (not close to rivers) were the principal areas in which black flies predominantly occurred. Some respondents (24.3%) answered that black flies bite throughout the region, and 4.0% claimed that they bite within houses. Most participants (75.0%) from Befang affirmed that flies always bite along rivers, while 41.2% from Mawong believed that farms were the preferred places where black flies were observed. In Soramboum, 2.3% had no idea of the times at which bites mostly occurred; however, 52.3% assumed that black flies bite during both the dry and rainy seasons, and 40.9% affirmed that black flies breed in fast-flowing water. In contrast, 41.2% of participants from Galim thought that black flies bite in the evening, 29.4% agreed that bites cause pruritus, and 26.5% assumed that black flies usually bite on farms. Generally, black flies were thought to bite at any time by 32.3%, in the morning by 19.0%, in the evening by 34.1%, and in the afternoon by 14.2% (Table 3). A significant difference was observed between our study areas (chi-square = 93.815, *p*-value < 0.001), where 10.6% of Befang reported that black flies acted in the morning and only 0.9% in Galim. However, 4.9% of Soramboum and 0.0% of Menchum reported that black fly bites could lead to blindness.

### 3.3. Knowledge of Community Respondents About Attractants to Black Flies

Four colors (white, black, red, and mixed) were included in the questionnaire. Of the respondents from Befang, 40.0% said that red was the most attractive color for flies, whereas 58.8% in Mawong considered all colors to attract black flies independently (Table 4).

In Soramboum and Galim, 54.5% and 72.1% of the participants thought that black flies bite independently of their body size, respectively. Regarding height, 8.8% of people in Galim believed that black flies mainly bite tall people, 76.5% in Mawong, 90.0% in Befang, and 70.5% in Soramboum, assuming that hosts are affected independently of their size and irrespective of their height. In Galim, 3.1% had no idea whether tall or short people were bitten.

### 3.4. Consolidation of All Data from the Investigated Regions

Below is a heatmap (Figure 3) consolidating the view of knowledge about onchocerciasis across four investigated regions. The data include responses to key factors related to blackfly behavior, breeding sites, and the effects of fly bites.

Befang has the strong awareness of blackfly activity in dry seasons and near rivers, but misconceptions about breeding sites. Galim recognizes fast-flowing water as a breeding site, which is correct. Soramboum has a more balanced understanding of seasons, preferred sites, and the effects of bites, but lacks specific knowledge on breeding. Mawong has some knowledge of biting behavior but incorrectly associates blackflies with malaria.

## 4. Discussion

In this study, we assessed the knowledge and practices in three localities in the Guinea savannah and one in the Sudan savannah. Our findings provide a scientifically relevant assessment of the knowledge and practices related to onchocerciasis along the Menchum Valley after 20 years of mass drug distribution and in two other localities (Galim and Soramboum) after 25–35 years of distribution [21]. The obtained data are crucial for evaluating the long-term impact of control strategies and identifying persistent knowledge gaps that may hinder disease elimination efforts.

A large proportion of residents were familiar with black flies, affirming their presence along riversides and corroborating established scientific findings that individuals working near rivers are at a heightened risk of *Simulium* biting and onchocerciasis transmission [30]. Our study also provides new insights into the community’s knowledge about black flies, an essential factor in designing culturally appropriate educational interventions. However, despite this familiarity with the vector, our study revealed significant deficiencies in community knowledge regarding disease causation, transmission, and prevention.

Only 9.2% of respondents correctly identified onchocerciasis as a consequence of black fly bites, while more than half mistakenly associated it with malaria. This highlights a major gap in disease awareness, potentially undermining prevention efforts and treatment adherence. Previous studies in African countries have documented variations in knowledge about onchocerciasis manifestations, with communities in high-prevalence areas demonstrating better awareness [24,25,26,27,28,29,30,31,32]. Our findings suggest that knowledge gaps persist despite decades of intervention, reinforcing the necessity of targeted educational campaigns. Notably, misconceptions about the disease were prevalent, with some respondents erroneously linking black fly bites to tuberculosis, and only 9.3% correctly identifying blindness as a major consequence of infection. This underscores the critical need for improved health education to enhance disease recognition and encourage early intervention.

A significant scientific contribution of this study is the documentation of persistent misconceptions regarding vector ecology. Only 17% of participants correctly identified fast-flowing rivers and streams as black fly breeding sites. This lack of knowledge presents a potential risk for ongoing transmission, as individuals may unknowingly engage in high-risk activities near these breeding grounds. Consistently with earlier reports from Cameroon and other African countries [20,33,34], some respondents (ranging from 4.5% in Soramboum to 55.0% in Menchum) incorrectly attributed black fly breeding to tree holes. These findings are scientifically relevant as they highlight an urgent need for community education on vector ecology to enhance control measures and reduce exposure.

The study also established a strong correlation between educational attainment and disease awareness, with those having higher levels of education demonstrating better knowledge and practices. In our study, 14.6% of participants were illiterate (primarily from Galim and Soramboum), and up to 50% had only attended primary school. This finding aligns with experience from Southwest Ethiopia, where illiteracy was also linked to poor knowledge of onchocerciasis [35]. The Vina du Nord Valley, where Soramboum is located, is one of the most isolated areas in the country, limiting access to education and health information. This geographic and socioeconomic barrier may contribute to the observed knowledge gaps and reinforces the importance of tailored educational interventions in remote communities.

Another key scientific insight from our study relates to seasonal variations in vector activity and community perceptions. While 61.1% of respondents correctly associated increased biting rates with the dry season, a discrepancy exists between this perception and previous studies in the savannah, which reported high monthly biting rates during the rainy season [36,37]. These discrepancies highlight the need for further entomological studies to refine vector control strategies based on local transmission patterns.

Our study also uncovered a lack of awareness regarding black fly attraction behavior, with more than 18% of respondents believing that white clothing attracts black flies. This finding is scientifically relevant as it informs community-based vector control strategies, such as personal protective measures [38,39].

The broader implication of our findings is that inadequate knowledge of onchocerciasis and its vector may compromise control efforts. Effective vector control campaigns must integrate scientific knowledge dissemination to ensure community engagement and behavioral change. Previous research has demonstrated that educational interventions significantly enhance knowledge and adherence to control measures [40,41]. Our study reinforces this by showing that persistent misinformation and inadequate awareness may contribute to the continued transmission of onchocerciasis despite decades of mass drug administration.

Public health interventions could use these data to target misconceptions (e.g., correcting malaria-blackfly confusion). Community education programs could focus on the role of fast-flowing water in breeding and pruritus/blindness as key symptoms. Prevention strategies (e.g., riverbank vegetation management, protective clothing) can be adapted based on where people believe blackflies are most active.

While this study provides valuable scientific insights, it has limitations. The absence of qualitative methods such as focus group discussions and epidemiological investigations limits the depth of understanding of community perceptions. Additionally, the study was conducted in only four rural communities, making it difficult to generalize the findings to all Cameroonian localities. Nevertheless, the data collection was rigorous, and the questionnaire was designed for clarity and reliability. Despite potential limitations in response options, our findings provide crucial evidence for guiding future intervention strategies.

In conclusion, our study highlights key deficiencies in knowledge and practices related to onchocerciasis, which may affect the effectiveness of mass drug distribution programs. Strengthening community awareness through targeted health education campaigns is essential to improving disease control efforts. Future research should expand to all affected localities and incorporate the operational aspects of drug distribution to enhance the effectiveness of onchocerciasis elimination programs. The scientific significance of this study lies in its contribution to understanding long-term community knowledge trends, informing evidence-based policy decisions, and optimizing intervention strategies for sustainable disease control.

## 5. Conclusions

Ultimately, this study offers significant fresh viewpoints on the knowledge and practices about *Simulium* black fly vector of onchocerciasis in Cameroonian endemic populations. Even with long-term mass ivermectin distribution, community knowledge of the disease’s spread still lags greatly, especially with regard to the part black fly vectors play. Although the majority of participants perceive the presence of black flies, many have misconceptions about the implications of their bites—for example, only a small percentage associate black fly bites with blindness, whereas a large number incorrectly correlate them with malaria. Additionally, the data show statistically significant regional variations in knowledge and practices, which are influenced by sociodemographic variables as age, occupation, and educational attainment. Misconceptions about vector ecology and disease transmission were especially noticeable in areas with lower literacy rates and less access to health education. These differences highlight the need for specialized, community-based educational initiatives that enhance vector surveillance, correct local misconceptions, and supplement current ivermectin distribution programs.

In general, effective onchocerciasis control and eventual elimination in these areas depend on raising public knowledge and dispelling myths through focused communication tactics. To improve these intervention techniques even more, future studies should include qualitative approaches and a wider geographic focus.

## Figures and Tables

**Figure 1 microorganisms-13-00736-f001:**
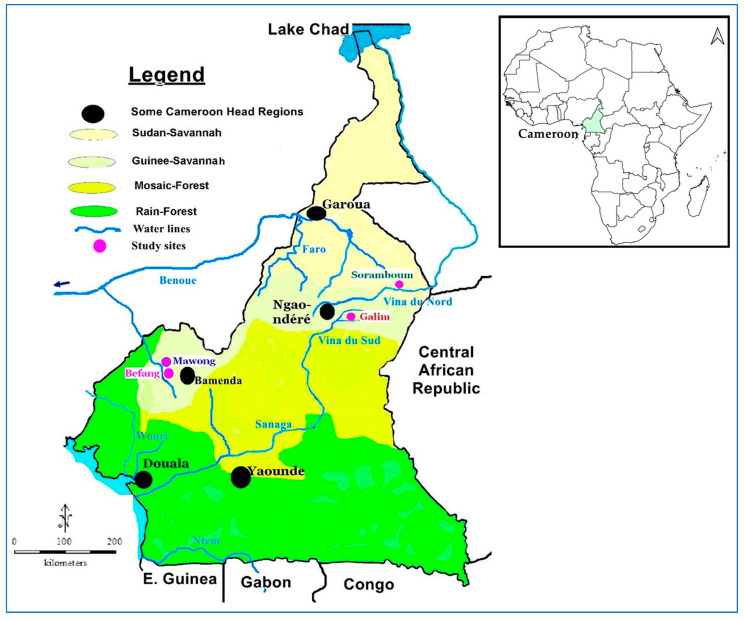
Map of Cameroon showing the study areas.

**Figure 2 microorganisms-13-00736-f002:**
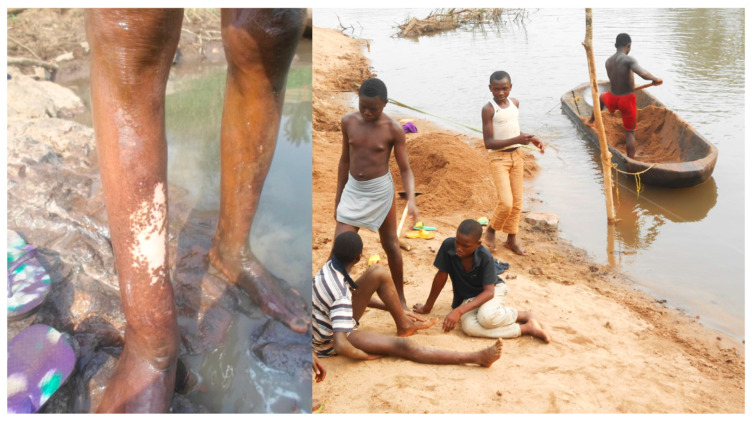
Personal observation during preliminary entomological survey in 2015: Leopard skin seen at the sand digging pool near the Menchum fall.

**Figure 3 microorganisms-13-00736-f003:**
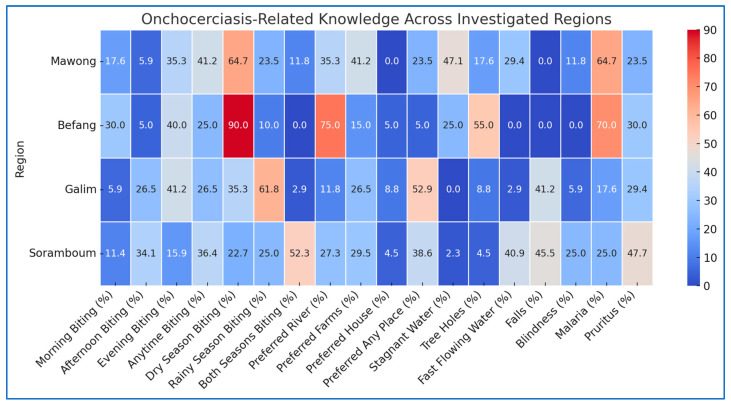
Heatmap of onchocerciasis-related knowledge across investigated regions. Red areas represent higher percentages, meaning a greater proportion of respondents in that region gave a specific answer; blue areas indicate lower percentages, suggesting fewer people identified that particular response; white/light shades represent intermediate values.

**Table 1 microorganisms-13-00736-t001:** Number of flies dissected by site and infection rate.

Capture Site	Number of Parous Flies Dissected (%)	Number of Flies Carrying *Onchocerca* Larvae (% of Parous Flies Infected)			
		L1	L2	L3	
				*O. v.*	*O. o.*
Befang	520 (61.3)	31 (5.96)	9 (1.73)	6 (1.15)	1 (0.19)
Mawong	38 (16.5)	2 (5.26)	0 (0.0)	4 (10.53)	0 (0.0)
Total	558 (51.8)	33 (5.91)	9 (1.61)	10 (1.79)	1 (0.18)

L1: number of parous flies carrying *Onchocerca* larvae stage 1 found; L2: carrying *Onchocerca* lavae stage 2; L3: carrying *Onchocerca* stage 3 (infective stage larvae); *O. v.*: *Onchocerca volvulus*; *O. o.*: *Onchocerca ochengi.*

**Table 2 microorganisms-13-00736-t002:** Sociodemographic characteristics of the study populations.

		Guinea Savannah			Sudan Savannah	*p*-Value
	Variables	Mawong N (%)	Befang N (%)	Galim N (%)	Soramboum N (%)	
Sex	Male	80 (58.8)	136 (85.0)	44 (64.7)	46 (52.3)	
	Female	56 (41.2)	24 (15.0)	24 (35.3)	42 (47.7)	<0.001
Age group	Less than 20	16 (11.8)	104 (65.0)	6 (8.8)	14 (15.9)	
	20 to 30	40 (29.4)	32 (20.0)	28 (41.2)	30 (34.1)	
	30 to 40	48 (35.3)	16 (10.0)	10 (14.7)	12 (13.6)	0.002
	40 to 50	32 (23.5)	8 (5.0)	24 (35.3)	32 (36.4)	
Occupation	School children	30 (22.1)	16 (10)	0 (0.0)	0 (0.0)	
	Sand dredging	2 (1.5)	94 (58.8)	0 (0.0)	0 (0.0)	
	Students	4 (2.9)	0 (0.0)	0 (0.0)	0 (0.0)	
	Farmers	56 (41.2)	32 (20.0)	20 (29.4)	80 (90.9)	
	Housekeeper	44 (32.3)	18 (11.3)	18 (26.5)	2 (2.3)	<0.001
	Carpenter	0 (0.0)	0 (0.0)	2 (2.9)	0 (0.0)	
	Motorcycle cab	0 (0.0)	0 (0.0)	8 (11.8)	0 (0.0)	
	Builder	0 (0.0)	0 (0.0)	2 (2.9)	0 (0.0)	
	Shepherd	0 (0.0)	0 (0.0)	10 (14.7)	3 (3.4)	
	Seamstress	0 (0.0)	0 (0.0)	2 (2.9)	0 (0.0)	
	Shopkeeper	0 (0.0)	0 (0.0)	2 (2.9)	0 (0.0)	
	Retired	0 (0.0)	0 (0.0)	2 (2.9)	3 (3.4)	
	Health worker	0 (0.0)	0 (0.0)	2 (2.9)	0 (0.0)	
Educational Status	Primary school	72 (52.9)	80 (50.0)	28 (41.2)	46 (52.3)	
	Secondary school	48 (35.3)	80 (50.0)	2 (2.9)	14 (15.9)	
	University	16 (11.8)	0 (0.0)	0 (0.0)	0 (0.0)	<0.0001
	Did not attend school	0 (0.0)	0 (0.0)	38 (55.9)	28 (31.8)	

%: percentage. The difference in the proportion of males and females between the Guinea Savannah and Soudan Savannah regions is statistically significant. The null hypothesis (H_0_) for the table assumes that there is no significant difference in the distribution of demographic characteristics (sex, age group, occupation, and educational status) across the four locations (Mawong, Befang, Galim, and Soramboum). Since the *p*-values are all < 0.001, the null hypothesis is likely rejected for all these variables, meaning that there are significant differences in sociodemographic characteristics between the locations.

**Table 3 microorganisms-13-00736-t003:** Comprehensive knowledge of community respondents regarding the biting activities of black flies.

		Guinea Savannah			Sudan Savannah	
	Variables	Mawong N (%)	Befang N (%)	Galim N (%)	Soramboum N (%)	*p*-Value
Biting period	Morning	24 (17.6)	48 (30.0)	4 (5.9)	10 (11.4)	<0.001
	Afternoon	8 (5.9)	8 (5.0)	18 (26.5)	30 (34.1)	
	Evening	48 (35.3)	64 (40.0)	28 (41.2)	14 (15.9)	
	Anytime	56 (41.2)	40 (25.0)	18 (26.5)	32 (36.4)	
	No idea	0 (0.0)	0 (0.0)	0 (0.0)	2 (2.3)	
Biting season	Dry season	88 (64.7)	144 (90.0)	24 (35.3)	20 (22.7)	<0.001
	Rainy season	32 (23.5)	16 (10.0)	42 (61.8)	22 (25.0)	
	Both	16 (11.8)	0 (0.0)	2 (2.9)	46 (52.3)	
Preferred biting site	River	48 (35.3)	120 (75.0)	8 (11.8)	24 (27.3)	<0.001
	Farms	56 (41.2)	24 (15.0)	18 (26.5)	26 (29.5)	
	House	0 (0.0)	8 (5.0)	6 (8.8)	4 (4.5)	
	Any place	32 (23.5)	8 (5.0)	36 (52.9)	34 (38.6)	
Breeding sites	Stagnant water	64 (47.1)	40 (25.0)	0 (0.0)	2 (2.3)	<0.001
	Tree holes	24 (17.6)	88 (55.0)	6 (8.8)	4 (4.5)	
	fast flowing water	40 (29.4)	0 (0.0)	2 (2.9)	36 (40.9)	
	Grass	0 (0.0)	0 (0.0)	2 (2.9)	0 (0.0)	
	Not known	8 (5.9)	32 (20.0)	30 (44.1)	6 (6.8)	
	Falls	0 (0.0)	0 (0.0)	28 (41.2)	40 (45.5)	
Preferred biting parts	Leg	48 (35.3)	152 (95.0)	0 (0.0)	4 (4.5)	<0.001
	Face	0 (0.0)	0 (0.0)	24 (35.3)	0(0.0)	
	Hands	0 (0.0)	0 (0.0)	4 (5.9)	4 (4.5)	
	Any exposed part	88 (64.7)	8 (5.0)	40 (58.8)	80 (91.0)	
Effect of fly bite	Blindness	16 (11.8)	0 (0.0)	4 (5.9)	22 (25.0)	<0.001
	Malaria	88 (64.7)	112 (70.0)	12 (17.6)	22 (25.0)	
	Pruritus	32 (23.5)	48 (30.0)	20 (29.4)	42 (47.7)	
	Tuberculosis	0 (0.0)	0 (0.0)	0 (0.0)	2 (2.3)	
	No idea	0 (0.0)	0 (0.0)	32 (47.1)	0 (0.0)	

The null hypothesis (H_0_) for this table states that there is no significant difference in the reported perceptions and experiences of fly-biting behaviors, breeding sites, and health effects across the four locations (Mawong, Befang, Galim, and Soramboum). Since the table shows *p*-values < 0.001 for all variables, we can conclude that the null hypothesis is rejected, indicating that the reported fly-biting patterns, breeding sites, and health effects differ significantly across the four locations.

**Table 4 microorganisms-13-00736-t004:** Proportions of respondents regarding biting preferences of black flies.

		Guinea Savannah			Sudan Savannah	
	Variables	Mawong N (%)	Befang N (%)	Galim N (%)	Soramboum N (%)	*p*-Value
Dress colour	White	32 (23.5)	24 (15.0)	12 (17.6)	14 (15.9)	<0.001
	Black	24 (17.6)	24 (15.0)	34 (50.0)	42 (47.7)	
	Red	0 (0.0)	64 (40.0)	4 (5.9)	6 (6.8)	
	All colours	80 (58.8)	48 (30.0)	18 (26.5)	26 (29.5)	
Body size	Fat	16 (11.8)	16 (10.0)	17 (25.0)	24 (27.3)	<0.001
	Slim	8 (5.9)	8 (5.0)	2 (2.9)	12 (13.6)	
	All	88 (64.7)	136 (85.0)	49 (72.1)	48 (54.5)	
	Not known	24 (17.6)	0 (0.0)	0 (0.0)	4 (4.5)	
Height	Short	0 (0.0)	0 (0.0)	108 (14.7)	10 (11.4)	<0.001
	Tall	8 (5.9)	16 (10.0)	6 (8.8)	14 (15.9)	
	All	104 (76.5)	144 (90.0)	38 (55.9)	62 (70.5)	
	Not known	24 (17.6)	0 (0.0)	14 (20.6)	2 (2.3)	

## Data Availability

The structured questionnaire is shown in Appendix A. All the data are available upon request.

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
