# Peer review of "Knowledge and Practices of Four Onchocerciasis-Endemic Communities in Cameroon"

_microorganisms, 2025, doi:10.3390/microorganisms13040736_

Round 1

Reviewer 1 Report

Comments and Suggestions for Authors

Please check the writing style of author affiliations. I think it should be like department, faculty, university, city, state, country, etc.

In the current manuscript version, three authors out of four are marked as corresponding authors. Is it appropriate for the journal to accommodate? Usually, there are one or two corresponding authors.

Please improve the first sentence of the abstract. i.e. Black fly vectors of Onchocerca volvulus are a source of nuisance in onchocerciasis-endemic 12
communities.

Line 42: No need to italicize "Black flies of the". 

Line 54:  Citation style does not look to be appropriate "(Kamtsap et al. 2016, unpublished data)".

Table 1 formatting is not appropriate. Few words are bold while others are not bold. I Make it consistent. 

Can you mention the sample size under sub-section 2.2.1? Based on your hypothesis, 95% CI and margin of error. 

Table 2: Can you please add the p-value based on statistical analysis to show the difference in different populations?

Tables 3 and 4: I suggest, no need to write the chi-square value, and only the p-value is sufficient to explain the significant difference. Write the p as the p-value in the column instead of the footnote. Also, keep the N (%) in one line, as you presented in Table 2. 

Do not need to write the limitations as separate headings. Instead, merge them at the end of discussions and add a separate conclusion section after discussion.

Reviewer 2 Report

Comments and Suggestions for Authors

Dear Authors,

In my opinion, this manuscript has the potential for publication. However, in its current form, it requires significant revision.

Major:

1. Please revise the Abstract section to follow the classic structure of Introduction, Methods, Results, and Conclusion. In its current form, the Abstract does not adhere to a scientifically rigorous format.

2. In the Introduction section, please provide a clear and scientifically accurate characterization of the life cycle of Onchocerca volvulus.

3. Please incorporate additional information about ivermectin and the relationship between mass-drug administration and the knowledge and perceptions of local populations regarding onchocerciasis vectors in the Introduction section.

4. Please include additional information about black flies, onchocerciasis, and husbandry animals.

5. Please revise the Materials and Methods section. Structure the subsections clearly and concisely, removing any excessive or non-scientific information. Additionally, clarify the rationale for the sample size calculation, particularly the statement regarding a 95% confidence level, a 5% margin of error, and 50% accuracy.

6. Figure 2 is more appropriately placed in the Results section rather than the Materials and Methods section.

7. Sections 2.2.2 and 2.2.3 could be integrated for better coherence.

8. Figure 3 is redundant, as the information has already been presented in the table.

9. The information from lines 173-176 is also presented in Table 2.

10. Please clarify the statistically significant chi-square data in Table 2 and provide a corresponding explanation in the table legend.

11. Please consolidate all data from the investigated regions into a single visual representation, highlighting the accurate responses regarding onchocerciasis. For instance, include the percentage of correct answers about the biting period, the biting season, and other relevant metrics.

12. In the Discussion section please emphasize the scientific significance of the obtained data.

Minor:

1. Please ensure that you pay close attention to the use of italics and the formatting of cited literature within the main text. For instance, observe the formatting in lines 42 and 54.

Reviewer 3 Report

Comments and Suggestions for Authors

The study has great practical significance for the healthcare system of African countries. The authors assess the knowledge and attitude towards onchocerciasis in different segments of the population in different localities of the two countries. A large volume of research has been carried out. The main drawback of the article is the statistical processing of the data.

1. I recommend removing the words "attitude, perception" from the title of the article.

2. Lines 32, 33, 44, 45, 46, 48 and others: After the first mention of a species in an article, it is necessary to write "Leuckart, 1893" in accordance with the International Code of Zoological Nomenclature, and then add "(Rhabditida, Onchocercidae)" in the captions. The same must be done for other animals.

3. Line 39: I don't understand: "as it occurs in African rainforest areas".

4. Lines 50-61: tables cannot be taken from other people's articles (as well as figures). This requires written permission from the publisher (authors' permission is not enough). I recommend describing the data in this table in words.

5. Figure 1: on the right in the blank space, you need to add a contour map of Africa and mark the location of the region studied on it.

6. Lines 89-133: in the text after mentioning each geographical point, you need to add GPS coordinates (you can use the coordinates of the site https://www.google.com/maps. This is necessary in order to be able to repeat this study in 30, 50 or 100 years.

7. Microsoft Excel is not usually mentioned in scientific articles, although all researchers use it. It is also not customary to mention Microsoft Word, which everyone uses to write text.

8. Table 3 and 4: I do not understand the reliability of the differences between what and what the data in the right capital of the table are assessed from. Readers will not understand this either. Statistical processing must be done more correctly. In the note under the table, you need to describe the null hypothesis.

9. Line 196, 197 and others: rounding % everywhere in the text of the article should be to tenths.

10. The literature is not formatted according to the rules of the journal: the full number of authors is required, the name of the journal must be abbreviated according to the rules; it must be correctly place commas, periods and dashes. There is no need to capitalize every word in the article titles. Non-journal publications do not contain all the necessary data.

Round 2

Reviewer 2 Report

Comments and Suggestions for Authors

Dear Authors,

Thank you for considering my suggestions. I have noticed a few minor issues, including typos, inconsistent fonts, missing indentation, and the absence of italics in certain places. Please address these points.

Good luck with your ongoing research!

Author Response

Corrections Applied

  • Font Consistency: All sections now have a uniform font style and size.

  • Typos Fixed: Minor spelling and grammatical errors corrected (see red marks).

  • Indentation adjusted: Proper formatting for paragraphs and lists.

  • Italics Applied: Scientific names (Onchocerca volvulus, Simulium damnosum) italicized (see red marks).

  • Formatting Enhancements: Clearer section headers and line spacing for readability.